# SenseFlow: A Physics-Informed and Self-Ensembling Iterative Framework for Power Flow Estimation

## Abstract

Power flow estimation plays a vital role in ensuring the stability and reliability of electrical power systems, particularly in the context of growing network complexities and renewable energy integration. However, existing studies often fail to adequately address the unique characteristics of power systems, such as the sparsity of network connections and the critical importance of the unique Slack node, which poses significant challenges in achieving high-accuracy estimations. In this paper, we present SenseFlow, a novel physics-informed and self-ensembling iterative framework that integrates two main designs, the Physics-informed Power Flow Network (PPFNet) and Self-ensembling Iterative Estimation (SeIter), to carefully address the unique properties of the power system and thereby enhance the power flow estimation. Specifically, SenseFlow enforces the PPFNet to gradually predict high-precision voltage magnitudes and phase angles through the iterative SeIter process. On the one hand, PPFNet employs the Virtual Node Attention and Slack-Gated Feed-Forward modules to facilitate efficient global-local communication in the face of network sparsity and amplify the influence of the Slack node on angle predictions, respectively. On the other hand, SeIter maintains an exponential moving average of PPFNet's parameters to create an ensemble model that refines power state predictions throughout the iterative fitting process. Experimental results demonstrate that SenseFlow outperforms existing methods, providing a promising solution for high-accuracy power flow estimation across diverse grid configurations[1].

## 1 Introduction

Power flow estimation is a crucial task for maintaining the stable and reliable operations of electrical power systems (Mhlanga, 2023; Khaloie et al., 2024). In practical power systems, any disturbance on a single bus can affect the overall balance of the system, necessitating a recalculation of the power flow to preserve stability. This makes power flow estimation not only essential but also highly frequent in operational contexts (Ngo et al., 2024). As shown in Figure 1(a), using the IEEE 39-bus system as an example, the network typically consists of three types of buses: multiple PQ (Load Bus) and PV (Generator Bus) nodes, and a single Slack node. The goal is to determine the voltage magnitudes and phase angles at each bus, adhering to the fundamental laws of power system dynamics. While traditional methods like Newton-Raphson (da Costa et al., 1999) and Gauss-Seidel (Eltamaly & Elghaffar, 2017) algorithms offer high accuracy, they face key challenges in modern power grids. As power networks grow in scale and complexity, especially with the integration of renewables, these mathematical solvers become computationally inefficient, particularly under large contingency (*e.g.*, N-K) analyses(Guo et al., 2021). Additionally, their reliance on complete parameters limits their applicability when critical information, such as nodes' reactive power, is missing or not measurable—an increasingly common issue in real-world scenarios (Hu et al., 2020).

In recent years, data-driven approaches, particularly deep learning techniques, have garnered significant attention for enhancing the accuracy and efficiency of power system analysis (Forootan et al., 2022). Among these approaches, Graph Convolutional Networks (GCNs) (Kipf & Welling, 2017)

---

[1]Code and logs are available in the supplementary materials.

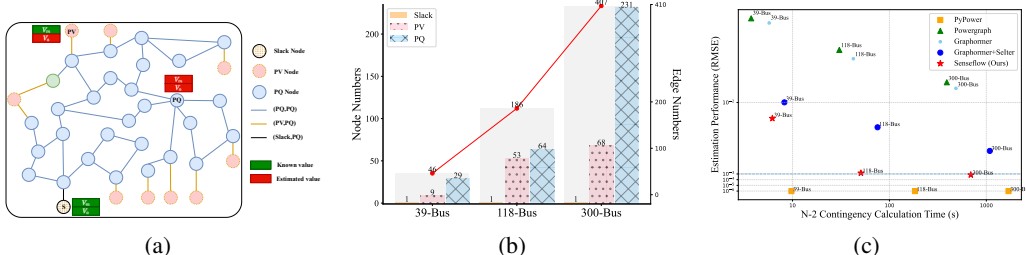

(a)                (b)                (c)

Figure 1: (a) Comparison of the number of nodes and edges across various IEEE standard systems (IEEE 39-Bus, 118-Bus, and 300-Bus), which reveals two key points: 1) there is only one slack node present in each system, and 2) the network exhibits relatively sparse connectivity. (b) Schematic diagram of the IEEE 39-Bus system with typical three different types of nodes and edges. The diagram also shows the parameters to be solved in the power flow calculation, including the phase angle of PV nodes and the voltage and phase angle of PQ nodes, alongside the known values including the voltage and phase angle of the slack node and the voltage of the PV nodes. (c) Tradeoff between the estimation performance on phase angles and the N-2 contingency calculation time.

have emerged as a prominent solution due to their effectiveness in handling graph-structured data, which aligns well with the inherent graph nature of power systems. However, despite their promise, many existing studies (Lin et al., 2024; Ngo et al., 2024; Hu et al., 2020) fall short of fully addressing the unique characteristics of power systems. As depicted in Figure 1(b), one of the key overlooked features is the presence of **only one single slack** bus in any size system, whose phase angle is used as a reference point for the entire system. The Slack bus is also the only node in the system that has both the known voltage magnitude and phase angle. Furthermore, power grids are fundamentally **sparse networks**: the number of edges typically scales linearly with the number of nodes (*i.e.*, $\mathcal{O}(N)$), which is considerably fewer than in fully connected graphs (*i.e.*, $\mathcal{O}(N^2)$). Such sparse connectivity limits information exchange between distant nodes, particularly concerning the Slack node, thereby posing a significant challenge for most GCN architectures that rely on graph connections for efficient node communication. To this end, we aim to employ physic-informed model designs that carefully integrate these distinct features to enhance power flow estimation.

On the other hand, most GCN-based methods follow an end-to-end fitting fashion (Lin et al., 2024; Nellikkath & Chatzivasileiadis, 2022; Falconer & Mones, 2022), which significantly enhances analytical efficiency by directly mapping input graphs to desired flow estimations. As shown in Figure 1(c), while this streamlined process enables rapid power flow analysis, such methods often sacrifice accuracy, as these models may not adequately capture the intricate dependencies and dynamics. In contrast, traditional power flow analysis methods (da Costa et al., 1999; Chang et al., 2007; Trias, 2012) typically employ iterative fitting techniques. These approaches gradually refine their predictions through successive approximations, improving the accuracy of voltage magnitudes and phase angles with each iteration. To this end, we aim to incorporate an iterative process into the GCN-based framework for a refined **tradeoff** between computational efficiency and high precision.

Inspired by these observed limitations, we propose a Physics-Informed and Self-Ensembling Iterative Framework for Power Flow Estimation, dubbed as **SenseFlow**, which seamlessly integrates two novel designs, the Physics-Informed Power Flow Network (PPFNet) and the Self-Ensembling Iterative Estimation (SeIter). **PPFNet** first adopts the Virtual Node Attention (VNA) module to aggregate the features of all nodes into a virtual node and apply cross-attention to distribute global information to individual PQ, PV, and Slack nodes. This facilitates efficient global-local communication without altering the original graph structure, ensuring that each node benefits from system-wide context. We also design the Slack-Gated Feed-Forward (SGF) module in PPFNet to emphasize the Slack node's significance by concatenating its features with PQ and PV nodes. A gated mechanism controls the Slack node's influence, while a residual connection preserves local node characteristics and enhances the Slack node's impact. **SeIter** guides PPFNet to iteratively predict changes in voltage magnitude and phase angle, gradually improving accuracy within each loop. During this process, an exponential moving average (EMA) of PPFNet's parameters maintains an ensemble model that generates more stable outputs, mitigating noise and fluctuations inherent in iterative training. Its

outputs are then fed into the next training loop, creating a self-ensembling cycle that progressively refines the predictions. In each loop, PPFNet is trained using two losses: the ground-truth loss to align predictions with actual voltage and phase values, and the equation loss to enforce adherence to power balance equations. Our main contributions are summarized as follows,

- We propose a novel power flow estimation framework, SenseFlow, which integrates two novel designs PPFNet and SeIter to obtain high-accuracy power flow estimation iteratively.
- Our PPFNet carefully addresses the unique characteristics of power systems by designing the Virtual Node Attention and Slack-Gated Feed-Forward modules, which enhance global-local communication and optimize the Slack node's influence effectively.
- Our SeIter strategy, equipped with a more stable and accurate self-ensembling model, progressively refines predictions to push the estimation into a high-precision space.
- Benefiting from the physic-informed design and iterative fitting strategy, our SenseFlow delivers leading performance in power flow estimation across different-size grid systems.

## 2 RELATED WORK

Power flow analysis is a fundamental task in electrical power systems that has been extensively researched for decades (Albadi & Volkov, 2020). Traditional methods, such as the Newton-Raphson Method (da Costa et al., 1999), Gauss-Seidel Method (Eltamaly & Elghaffar, 2017), and Backward-Forward Sweep (Chang et al., 2007), provide promising estimation accuracy through iterative optimization procedures. However, these methods often struggle to scale effectively with larger and more complex power systems, particularly those incorporating renewable energy sources (Ngo et al., 2024). Consequently, research groups have increasingly shifted their focus towards data-driven approaches (Forootan et al., 2022; Khaloie et al., 2024; Goodfellow et al., 2016). Studies along this line aim to fit the distribution of the collected historical data or simulated data for accurate and efficient power flow approximation. Considering the collinearity of the training data and the nonlinearity of the power flow model, Chen et al. (2021) proposes a piecewise linear regression algorithm for model fitting. Similarly, Guo et al. (2021) converts the nonlinear relationship of flow calculation into a higher dimension state space based on the Koopman operator theory. However, most of these works focused on the nonlinear fitting ability of the model and ignored the graph-structured topology nature of power systems, leading to unsatisfying estimation performance.

Graph Convolutional Networks (GCNs) (Wu et al., 2020; Zhang et al., 2020) are powerful models designed to handle graph-structured data and have demonstrated significant potential in addressing the graph topology in power systems(Liao et al., 2021; Falconer & Mones, 2022; Lopez-Garcia & Domínguez-Navarro, 2023). The work by Owerko et al. (2020) highlights the promising capability of GCN to leverage the network structure of the data and approximates a specified optimal solution through an imitation learning framework. Recent studies have incorporated the physical constraints of power systems into the loss design, enhancing estimation accuracy and robustness to the variations of typologies (Lin et al., 2024; Gao et al., 2023; Hu et al., 2020). For instance, Habib et al. (2023) adopts a weakly supervised learning method based on power flow equations, which removes the requirement for labeled data but results in relatively lower accuracy than fully supervised approaches. PowerFlowNet (Lin et al., 2024) introduces a joint modeling approach that simultaneously represents both buses and transmission lines, conceptualizing power flow estimation as a GNN node-regression problem. However, none of these studies thoroughly examine the distinctive characteristics of power systems, such as network sparsity and the critical role of the slack node. Differently, we explore these features and deliberately incorporate them into our network designs.

## 3 SENSEFLOW

### 3.1 OVERVIEW

Given a power system network $\mathcal{G}$ with $N$ buses (nodes) and $E$ transmission lines (edges), the objective of power flow estimation is to determine the voltage magnitudes $V_{m,i}$ and phase angles $V_{a,i}$ at each bus $i \in \{1, 2, \ldots, N\}$, subject to the power balance equations that govern active and reactive power flows in the network. In terms of the training process, we have the active/reactive power for

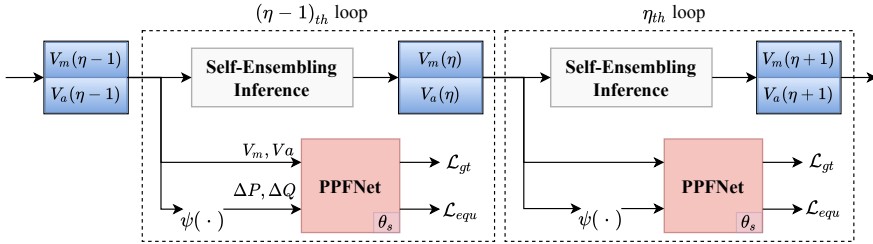

(a) Iterative power flow estimation

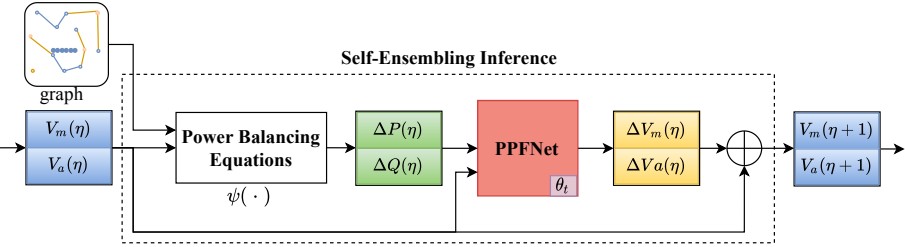

(b) Self-ensembling prediction for the subsequent loop

Figure 2: Illustration of our Self-ensembling Iterative Estimation (SeIter). **(a)** In the $\eta$-th loop, the trainable PPFNet ($\theta_s$) receives the input voltage magnitudes $V_m(\eta)$ and phase angles $V_a(\eta)$ from the previous loop and the changes in active and reactive power $\Delta P$ and $\Delta Q$ calculated by the power balancing equations $\psi$. The net is trained by two loss functions: the ground-truth loss, $\mathcal{L}_{gt}$, which aligns the predictions with the actual data, and the equation loss, $\mathcal{L}_{equ}$, which ensures the model adheres to the physical laws governing the system. **(b)** The Self-Ensembling Inference module prepares the updated data for the next loop. It leverages the self-ensembling teacher model ($\theta_t$) to generate predictions, which serve as the input for the trainable model in the subsequent $\eta + 1$ loop, where $\theta_t$ is updated by the exponential moving averaging of $\theta_s$.

PQ nodes, *i.e.*, $P^{\mathrm{PQ}}/Q^{\mathrm{PQ}}$, active power $P^{\mathrm{PV}}$ and voltage magnitude $V_m^{\mathrm{PV}}$ for PV nodes, and known voltage $V_m^{\mathrm{Slack}}$ and phase angle $V_a^{\mathrm{Slack}}$ for the Slack node, as well as the network topology encoded in the admittance matrix. Giving the ground-truth information on the PQ and PV nodes, including $V_m^{\mathrm{PQ}}, V_a^{\mathrm{PQ}}, V_a^{\mathrm{PV}}$, our goal is to obtain corresponding accurate predictions.

Our proposed SenseFlow framework addresses the power flow estimation problem by seamlessly integrating physics-informed modeling with a self-ensembling iterative learning process. At its core, SenseFlow leverages both the unique structural features of power systems and the iterative refinement capabilities of ensembling models. Specifically, SenseFlow trains the proposed PPFNet via the SeIter strategy. PPFNet process input data $\mathcal{G}(N, E)$ with known features, $P^{\mathrm{PQ}}, Q^{\mathrm{PQ}}), P_{\mathrm{PV}}, V_m^{\mathrm{PV}}, V_m^{\mathrm{Slack}}, V_a^{\mathrm{Slack}}$ to predict the unknown values on the PV and PQ nodes, *i.e.*, the voltage magnitude $\hat{V}_m^{\mathrm{PQ}}$ and phase angle $\hat{V}_a^{\mathrm{PQ}}$ for the PQ nodes, and phase angle $\hat{V}_a^{\mathrm{PV}}$ for the PV nodes. The training of PPFNet is guided by a ground-truth loss $\mathcal{L}_{g}t$, and the power balancing equation loss $\mathcal{L}_{equ}$,

$$\mathcal{L} = \mathcal{L}_{gt} + \lambda \mathcal{L}_{equ}, \qquad (1)$$

where $\lambda$ is a scalar hyper-parameter to adjust the equation loss weight. Similar to Lopez-Garcia & Domínguez-Navarro (2023); Hu et al. (2020), we use L1 loss for the ground-truth supervision,

$$\mathcal{L}_{gt} = \frac{1}{N_{\mathrm{PQ}}} \sum_{i=1}^{N_{\mathrm{PQ}}} \left( \left| \hat{V}_{m,i}^{\mathrm{PQ}} - V_{m,i}^{\mathrm{PQ}} \right| + \left| \hat{V}_{a,i}^{\mathrm{PQ}} - V_{a,i}^{\mathrm{PQ}} \right| \right) + \frac{1}{N_{\mathrm{PV}}} \sum_{j=1}^{N_{\mathrm{PV}}} \left| \hat{V}_{a,j}^{\mathrm{PV}} - V_{a,j}^{\mathrm{PV}} \right|. \qquad (2)$$

The power balancing equation loss is applied to encourage minimal power changes,

$$\mathcal{L}_{equ} = \frac{1}{N_{\mathrm{PQ}}} \sum_{i=1}^{N_{\mathrm{PQ}}} \left( \left| \Delta P_i^{\mathrm{PQ}} \right| + \left| \Delta Q_i^{\mathrm{PQ}} \right| \right) + \frac{1}{N_{\mathrm{PV}}} \sum_{j=1}^{N_{\mathrm{PV}}} \left| \Delta P_j^{\mathrm{PV}} \right|, \qquad (3)$$

where the calculations of $\Delta P$ and $\Delta Q$ are involved in the SeIter process. Through the SeIter strategy, SenseFlow refines its predictions by iteratively updating voltage magnitudes and phase angles. A self-ensembling mechanism, maintained by exponential moving averages, ensures stability during the iterative process, progressively pushing the predictions toward higher accuracy. We will detail these two main designs in the following sections.

## 3.2 SELF-ENSEMBLING ITERATIVE ESTIMATION

The self-ensembling iterative estimation (SeIter) diverges from conventional end-to-end learning approaches. Instead of directly fitting inputs to final voltage magnitudes and phase angles, SeIter gradually enforces the trainable module to approach the ground truth with the help of a self-ensembling prediction. As shown in Figure 2(a), the trainable model focuses on fitting the incremental changes in voltage and phase angle, allowing for refined adjustments with each cycle. This iterative refinement enables the model to achieve accuracy levels that end-to-end approaches may not reach.

In the SeIter, each iteration, denoted as the $\eta$th loop, involves a dual approach that focuses on both training the PPFNet model and refining the estimates for voltage magnitude and phase angle for the future loop. On the one hand, as shown in Figure 2(a), the input data is first utilized to train the PPFNet, parameterized by $\theta_s$, by minimizing the ground truth loss $\mathcal{L}_{gt}$. Second, the input data is subjected to the power balance equations, which yield incremental changes in active power $\Delta P$ and reactive power $\Delta Q$. The objective here is to minimize the equation loss $\mathcal{L}_{equ}$, which is designed to ensure that the total power variations approach zero. Let $\psi(V_m, V_a, \mathcal{G})$ denote the Power balancing equations, the active and reactive power changes $\Delta P_i$ and $\Delta Q_i$ at the bus $i$ can be calculated by,

$$\Delta P_i = P_i - \sum_{j=1}^{N} |V_{m,i}||V_{m,j}|(G_{ij}\cos(V_{a,i} - V_{a,j}) + B_{ij}\sin(V_{a,i} - V_{a,j})), \quad (4)$$

$$\Delta Q_i = Q_i - \sum_{i=1}^{N} |V_{m,i}||V_{m,j}|(G_{ij}\sin(V_{a,i} - V_{a,j}) - B_{ij}\cos(V_{a,i} - V_{a,j})), \quad (5)$$

where $G_{ij}$ and $B_{ij}$ represent the conductance and susceptance of the line connecting buses $i$ and $j$.

On the other hand, as shown in Figure 2(b), the input data is processed through the Self-Ensembling Inference module, which maintains an ensembling model, parameterized by $\theta_t$, updated by exponential moving averaging (EMA) of the PPFNet parameters, *i.e.*,

$$\theta_t \leftarrow \alpha\theta_t + (1 - \alpha)\theta_s, \quad (6)$$

where $\alpha$ is a common momentum parameter. The ensembling model acts as a stable reference point, providing an output that reflects the accumulated knowledge from the iterative training process. Its output is further used as the input for the subsequent iteration, *i.e.*, the $(\eta + 1)$th loop. This self-ensembling iterative estimation allows the trainable model to benefit from the progressively refined outputs of the ensembling model, thereby enhancing its learning capabilities and improving the overall convergence of the solution.

## 3.3 PHYSICS-INFORMED POWER FLOW NETWORK

As shown in Figure 3, our proposed PPFNet is built upon two fundamental modules: the Virtual Node Attention (VNA) and Slack-Gated Feed-Forward (SGF). VNA enables each node to perceive global changes without disrupting the underlying graph structure, while SGF enhances the influence of the slack node on each PQ and PV node, fostering accurate phase angle predictions.

*Virtual Node Attention.* Our VNA is specifically designed to address the sparsity issue by providing each node with the ability to sense and respond to global system variations. This design ensures that each local node can dynamically adjust its state in response to changes in the overall system, thus accurately capturing the interdependencies that are essential for maintaining the stability and reliability of power systems. By incorporating the VNA, we enable a more comprehensive and adaptive modeling of global interactions, ensuring that the system-wide impact of local changes is appropriately reflected. Specifically, We obtain the virtual node representation by contacting all the

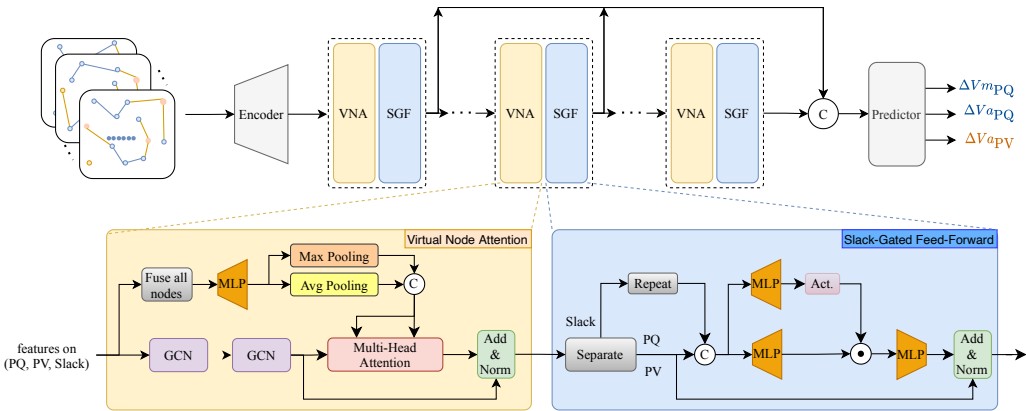

Figure 3: Illustration of our proposed PPFNet, which mainly consists of two main modules, the Virtual Node Attention (VNA) and Slack-Gated Feed-Forward (SGF). THe whole hetero-graph is fed into the network. The VNA creates a virtual node by combining and pooling the features of all nodes, then uses cross-attention to selectively communicate global information to each node type. This enhances the interaction between global and local information, preserving the graph structure while improving the model's ability to capture system-wide dependencies. The SGF combines the slack node's features with each node's features through a gated feed-forward network, enhancing the slack node's influence on other nodes while preserving the original node characteristics via a residual connection. Best viewed on screen.

node features $F_{\text{PQ}}, F_{\text{PV}}, F_{\text{Slack}}$ without breaking the graph structure,

$$F_{\text{afuse}} = \text{Linear}(\text{Concat}(F_{\text{PQ}}, F_{\text{PV}}, F_{\text{Slack}})) \tag{7}$$

$$F_{\text{Vnode}} = \text{Concat}(AvgPool(F_{\text{afuse}}), MaxPool(F_{\text{afuse}})) \tag{8}$$

Meanwhile, we can obtain the updated node representation after the graph neural network,

$$F_{\star} = \text{GCN}(F_{\star}), \star \in \{\text{PQ, PV, Slack}\} \tag{9}$$

where GCN denotes multi-layer graph convolutional network (*e.g.*, GraphConv (Morris et al., 2019), GAT (Veličković et al., 2018)). Subsequently, we attend the global information to each type of the power node via the cross attention,

$$F_{\star} = \text{LayerNorm}\left(F_{\star} + \text{softmax}\left(\frac{F_{\star} \cdot F_{\text{Vnode}}^T}{\sqrt{d_k}}\right) F_{\text{Vnode}}\right) \tag{10}$$

where $d_k$ is the dimension of the $F_{\text{Vnode}}$ vectors. In this way, our VNA module preserves the original graph structure and bridges the connection between each node and the whole system without implicitly introducing auxiliary nodes and edges.

*Slack-Gated Feed-Forward.* Our SGA effectively enhances the influence of the slack node in power system modeling by concatenating its feature representation with the feature representations of each PQ or PV node. The combined features are then processed through a gated feed-forward network, allowing the slack node's influence to be dynamically adjusted based on the current state of the node. Moreover, a residual connection is added, incorporating the original node features to ensure that local characteristics are preserved while enhancing the model's ability to accurately capture phase angle relationships throughout the system. Taking the PV node as an example, we have,

$$F_{\text{sfuse}} = \text{Linear}(\text{Concat}(F_{\text{PQ}}, F_{\text{Slack}})) \odot \sigma(\text{Linear}(\text{Concat}(F_{\text{PQ}}, F_{\text{Slack}}))) \tag{11}$$

$$F_{\text{PQ}} = \text{LayerNorm}(F_{\text{PQ}} + \text{Linear}(F_{\text{sfuse}})). \tag{12}$$

To construct the complete model, as shown in Figure 3, we stack $K$ layers of these blocks, allowing for deeper feature extraction and representation learning. In the end, the outputs from all blocks are concatenated and then fed into a predictor module to predict the voltage and phase angles.

Table 1: Performance comparison on the IEEE 39-Bus and IEEE 118-Bus system in terms of the root mean squared error (RMSE), where lower values indicate better performance. "+SeIter" indicates the application of our proposed self-ensembling iterative estimation process to the corresponding method. The best results are highlighted in **bold**.

| Method | IEEE 39-Bus | | | IEEE 118-Bus | | |
|---|---|---|---|---|---|---|
| | $PQ_{Vm}$ | $PQ_{Va}$ | $PV_{Va}$ | $PQ_{Vm}$ | $PQ_{Va}$ | $PV_{Va}$ |
| PowerGraph (Varbella et al., 2024) | 0.01390720 | 0.30019834 | 0.32490083 | 0.00822813 | 0.10501729 | 0.11086510 |
| PowerflowNet (Lin et al., 2024) | 0.00700982 | 0.09310086 | 0.09836672 | 0.00188223 | 0.02760057 | 0.02899146 |
| TGN (Lopez-Garcia & Domínguez-Navarro, 2023) | 0.00489032 | 0.05125980 | 0.053372381 | 0.00138807 | 0.01697165 | 0.01895702 |
| GraphConv | 0.00724108 | 0.10125969 | 0.12637231 | 0.00192112 | 0.03769957 | 0.03597135 |
| GINEConv | 0.00768264 | 0.10450818 | 0.12893821 | 0.00194470 | 0.03934504 | 0.03760612 |
| SageConv | 0.00755344 | 0.10445687 | 0.12901593 | 0.00192444 | 0.04449241 | 0.04275129 |
| ResGatedGraphConv | 0.00694495 | 0.10085707 | 0.12677170 | 0.00130103 | 0.03659180 | 0.03513782 |
| GatConv | 0.00808900 | 0.10591513 | 0.13207403 | 0.00262339 | 0.04434326 | 0.04388360 |
| TransformerConv | 0.00722702 | 0.10429660 | 0.13010464 | 0.00147067 | 0.04356860 | 0.04153621 |
| FlowNet (**ours**) | 0.00453724 | 0.04653547 | 0.05373371 | 0.00115526 | 0.01273561 | 0.01269017 |
| + SeIter (*i.e.*, **SenseFlow**) | **0.00078161** | **0.00608600** | **0.00609802** | 0.00009817 | **0.00102664** | **0.00103545** |

## 4 EXPERIMENT

### 4.1 DATASETS

We construct our dataset based on standard IEEE test cases (39-Bus, 118-Bus, and 300-Bus) using Matpower (Zimmerman et al., 2010), following approaches similar to Lopez-Garcia & Domínguez-Navarro (2023) and Gao et al. (2023). To simulate diverse scenarios, we introduce variations in power injections, branch characteristics, and grid topology. Specifically, we apply uniform noise to the active and reactive power loads ($P$ and $Q$), adjusting them to range between 50% and 150% of their original values. Likewise, branch features are perturbed with uniform noise, ranging from 90% to 110% of their baseline values. To examine different grid topologies, we randomly disconnect one or two transmission lines in each sample. All load bus voltage magnitudes are initialized at 1 P.U., and phase angles are set relative to the slack bus reference angle. In this way, we generate 100,000 samples for the 39-Bus and 118-Bus systems, and 500,000 samples for the 300-Bus system. 20% of the records are reserved as test sets, with strictly distinct grid topologies from the training data.

### 4.2 IMPLEMENTATION DETAILS

In our experiments, we utilized a batch size of 256 and employed the Adam optimizer with a learning rate set at 0.001, which follows a cosine decay schedule down to 1e-5 over a total of 100 epochs. Regarding feature embedding sizes, we set them to 128 for the IEEE 39-Bus and 118-Bus systems, while a size of 256 was used for the IEEE 300-Bus system. To effectively integrate information, we stacked a block that combines Virtual Node Attention and Slack-Gated Feed-Forward modules a total of four times. Our models are trained and inferred using an iterative fitting approach with 8 loops to enhance the estimation accuracy. All code was implemented in PyTorch 2.1, and both training and testing were conducted on the 40GB A100 GPU.

We evaluated our approach against recent studies like Powergraph (Varbella et al., 2024) (with the best transformer-based solution), PowerFlowNet (Lin et al., 2024), and TGN (Lopez-Garcia & Domínguez-Navarro, 2023), and also popular graph networks commonly used in power system analysis, including GraphConv, GINEConv, SageConv, ResGatedGraphConv, GatConv, and TransformerConv. The metrics for comparison focused on the root mean square error of voltage and phase angle predictions for PQ nodes, as well as phase angle predictions for PV nodes.

### 4.3 ESTIMATION PERFORMANCE

Table 1 presents a performance comparison between our proposed method, SenseFlow (comprising PPFNet and SeIter), and other advanced approaches on the IEEE 39-Bus and IEEE 118-Bus systems. The results demonstrate that SenseFlow significantly outperforms the other methods across both systems. In the IEEE 39-Bus system, SenseFlow achieves the lowest root mean square error (RMSE)

Table 2: Performance comparison on IEEE 300-Bus system. All notations are the same as in Table 1.

| Method | w/o SeIter | | | w/ SeIter | | | Param. |
| --- | --- | --- | --- | --- | --- | --- | --- |
| | PQ$_{Vm}$ | PQ$_{Va}$ | PV$_{Va}$ | PQ$_{Vm}$ | PQ$_{Va}$ | PV$_{Va}$ | |
| GraphConv | 0.00088801 | 0.01430215 | 0.01519640 | 0.00018706 | 0.00177910 | 0.00158296 | 8.422M |
| GINEConv | 0.00086362 | 0.01507135 | 0.01591485 | 0.00022936 | 0.00213723 | 0.00190035 | 4.227M |
| SageConv | 0.00091070 | 0.01600684 | 0.01706942 | 0.00025046 | 0.00243048 | 0.00223354 | 8.422M |
| ResGatedGraphConv | **0.00051189** | 0.01318974 | 0.01419000 | 0.00013985 | 0.00147823 | 0.00124201 | 17.105M |
| GatConv | 0.00291205 | 0.01372243 | 0.02828574 | 0.00039533 | 0.00292789 | 0.00362555 | 34.112M |
| TransformerConv | 0.00053179 | 0.01439258 | 0.01582433 | 0.00016199 | 0.00206462 | 0.00216299 | 55.083M |
| SenseFlow (**ours**) | 0.00093000 | **0.00417808** | **0.00473750** | **0.00010600** | **0.00086501** | **0.00077378** | 21.844M |

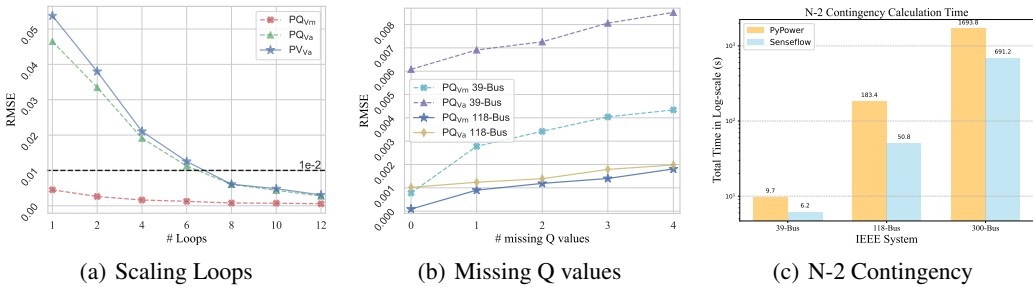

(a) Scaling Loops  (b) Missing Q values  (c) N-2 Contingency

Figure 4: (a) we examine the impact of the iterative loop on the IEEE 39-Bus system. The number of iterations is set to 8 by default, considering the increased inference effort with larger loops. (b) we investigate the missing Q values settings on IEEE 39-Bus and 118-Bus. These settings, without complete known information, cannot be addressed by conventional calculation methods. (c) we compare the total calculation time of N-2 contingency analysis on IEEE standards.

for voltage predictions at PQ and PV nodes, with magnitude error at 0.0007816 and phase angle errors at 0.00608600 and 0.00609802, showcasing its high-precision predictive capabilities. In the IEEE 118-Bus system, SenseFlow also exhibits exceptional performance. While it may not be the absolute best for magnitude predictions of PQ nodes, it remains very competitive and shows remarkable superiority in the more challenging phase angle predictions compared to other methods. Overall, the combination of the PPFNet architecture and the SeIter strategy positions SenseFlow as a highly effective approach for power system state estimation.

We investigate the estimation performance of our SenseFlow on the more complex and larger IEEE 300-Bus system in Table 2. We can clearly observe that our SenseFlow with SeIter can obtain the state-of-the-art (SOTA) performance, evidenced by consistently lower RMSE values across different metrics. In the absence of our SeIter strategy, we achieved a significant reduction in phase angle prediction error from approximately 0.013 to around 0.004 compared to the second-best method, ResGatedGraphConv. When SeIter is incorporated, SenseFlow emerges as the only method capable of reducing phase angle errors below 1e-3, showcasing its superior performance in this context. These improvements highlight the effectiveness of our method in handling complex power system scenarios and underscore its potential for real-world applications.

We reproduce recent studies like Powergraph on our more challenging datasets with larger perturbations and varying topologies, and find they fail to handle such settings. Notably, our method performs increasingly better on large 118 and 300-bus systems, since our test sets are built from N-2/3 topology perturbations on IEEE systems, which affect the 39-bus most and make it the hardest.

## 4.4 ABLATION STUDY

*Impact of different components of SenseFlow.* The ablation studies in the Table 3 demonstrate the effectiveness of the key components in our SenseFlow, including the self-ensembling iterative estimation (SeIter), block fusion, Virtual Node Attention (VNA), and Slack-Gated Feed-Forward (SGF).

Table 3: Ablation studies on our SenseFlow. We examine the effectiveness of the self-ensembling iterative estimation process (SeIter) and the main components of our proposed FlowNet, including the block fusion, Virtual Node Attention (VNA) and Slack-Gated Feed-Forward (SGF). Results are reported on the IEEE 39-Bus. Improvements over the baseline are marked in blue.

| SeIter | FlowNet | | | | RMSE ↓ | | |
|---|---|---|---|---|---|---|---|
| | Base | Fusion | VNA | SGF | $PQ_{Vm}$ | $PQ_{Va}$ | $PV_{Va}$ |
| | ✓ | | | | 0.00914456 | 0.12542769 | 0.14066372 (0.0) |
| | ✓ | ✓ | | | 0.00774126 | 0.10663362 | 0.12872563 (↓ 0.01193809) |
| | ✓ | ✓ | ✓ | | 0.00561620 | 0.05007443 | 0.05717816 (↓ 0.08348556) |
| | ✓ | ✓ | | ✓ | 0.00658822 | 0.07125929 | 0.07577518 (↓ 0.06488854) |
| | ✓ | ✓ | ✓ | ✓ | **0.00453724** | **0.04653547** | **0.05373371** (↓ 0.08693001) |
| ✓ | ✓ | | | | 0.00102207 | 0.01159813 | 0.01238586 (0.0) |
| ✓ | ✓ | ✓ | | | 0.00112893 | 0.01129249 | 0.01206334 (↓ 0.00032252) |
| ✓ | ✓ | ✓ | ✓ | | 0.00098343 | 0.00697184 | 0.00771528 (↓ 0.00467058) |
| ✓ | ✓ | ✓ | | ✓ | 0.00100311 | 0.01011067 | 0.01089543 (↓ 0.00149043) |
| ✓ | ✓ | ✓ | ✓ | ✓ | **0.00078161** | **0.00608600** | **0.00609802** (↓ 0.00628784) |

Without the SeIter process, introducing the Fusion, VNA, and SGF results in RMSE reductions of 0.01193809, 0.08348556, and 0.06488854, respectively, for the phase angle predictions of PV nodes compared to the baseline. When these components are combined, forming the complete PPFNet, the RMSE is further reduced to 0.0537, an overall improvement of 0.0869. More notably, the addition of SeIter (with a default loop count of 8) significantly decreases all RMSE metrics by approximately 10-fold. As a result, our complete SenseFlow achieves an RMSE of less than 1e-3 for the voltage magnitude estimation and less than 1e-2 for the phase angle estimation, demonstrating its substantial improvements and overall effectiveness.

*Scaling iterative loops.* Figure 4(a) investigate the effect of scaling iterative loops on the estimation performance. Specifically, transitioning from a single loop to multiple loops significantly enhances the accuracy of voltage magnitude and phase angle predictions, with up to 12 loops reducing the phase angle error by nearly two orders of magnitude. As the number of loops increases, prediction errors continue to decrease, highlighting the benefits of iterative refinement. However, this improvement comes at the cost of increased training and inference costs. To balance accuracy and computational efficiency, we adopt 8 loops as the default, which ensures a phase angle prediction error below 1e-2 while minimizing computational overhead.

*Comparison with mathematical methods.* **Calculation time.** We compare the total time required to perform full N-2 contingency simulations on three IEEE test systems using PyPower (a traditional solver) and our Senseflow ( on single GPU-A100-40G). While performance is similar for small systems, as shown in Figure 4(c), Senseflow achieves 3–5 speedup on larger networks. This demonstrates the model's superior scalability for high-volume, large-scale contingency analysis. **Estimation with incomplete inputs.** SenseFlow is capable of accurately estimating voltage states with missing parameters, a challenge that conventional methods cannot address due to incomplete information. As shown in Figure 4(b), with missing inactive power in PQ nodes, our method still achieves estimation performance at the $10^{-3}$ level, even when more than 10% of PQ nodes lack Q-values on the 39-Bus system. Compared with TransformerConv and GatConv (Table 1), SenseFlow achieves lower errors on the 118-Bus system even with Q-value missing.

## 5 CONCLUSION

In this paper, we emphasize the importance of the unique features of power systems for power flow analysis, specifically the sole phase angle-referencing Slack node and the sparse network structure. To this end, we propose SenseFlow, a novel Physics-Informed and Self-Ensembling Iterative Framework for power flow estimation. By integrating the proposed PPFNet and SeIter strategy, SenseFlow effectively addresses these characteristics and further enhances the prediction accuracy of voltage magnitudes and phase angles through iterative refinement. Experimental results demonstrate that our SenseFlow achieves leading performance in power flow estimation, and extensive ablation studies validate the effectiveness of the proposed components and strategies.

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
