# OpenReview forum: "SenseFlow: A Physics-Informed and Self-Ensembling Iterative Framework for Power Flow Estimation"
_ICLR.cc/2026/Conference — ICLR 2026 Conference Withdrawn Submission_

### Official Review · Reviewer_dVzt · 2025-10-22

**Soundness:** 3
**Presentation:** 3
**Contribution:** 3
**Rating:** 6
**Confidence:** 3

**Summary:**

The authors propose SenseFlow that combines two components designed in this work: PPFNet and SeIter. PPFNet (Physics-informed Power Flow Network) includes Virtual Node Attention (VNA) and Slack-Gated Feed-Forward (SGF) modules, where VNA condense all nodes into a virtual node to distill global information and distribute the information through cross-attention; SGF emphasizes the slack node (only one slack node in each dataset) by combining its features with each node's features. SeIter (Self-Ensembling Iterative Estimation) is an iterative strategy that refines the PPFNet estimations for multiple loops to approach ground truth.

**Strengths:**

1. This work targets power flow estimation, achieving significantly higher estimation quality compared to neual-network-based baselines, and lower estimation time compared to traditional numerical solver (PyPower).
2. This work covers the scenario of missing data in a power system, which is difficult to handle in numerical solvers.
3. The domain specific technologies proposed in this work are reasonable and effective as experimental results demonstrate.

**Weaknesses:**

1. The motivation may be more clearly stated.
2. The impact of this work may be generalized to other domains.
3. Missing robustness discussion.

**Questions:**

1. In practice, why is the speed of power flow estimation important? Is it for timely prediction and reaction to avoid, e.g., cascading failure? What level of estimation latency is acceptable? Can SenseFlow achieve that level of latency?
2. Can this work be generalized to other domains?
3. As real-world power systems are affected by noise, could you discuss how the network handles noise?
4. The iterative approach reminds me of Neural ODE, which interprets ResNet as a discrete step of an ODE. The paper also uses iterative evaluations to fit the trajectory of an ODE. Considering a power grid is governed by differential equations in theory, it would be insightful if the authors can discuss theoretically if SenseFlow can be accommodated into the frame of Neural ODE -- is it a neural-network-based differential equation solver?

Chen, Ricky TQ, et al. "Neural ordinary differential equations." Advances in neural information processing systems 31 (2018).

---

### Official Review · Reviewer_9gpy · 2025-10-30

**Soundness:** 2
**Presentation:** 1
**Contribution:** 2
**Rating:** 2
**Confidence:** 5

**Summary:**

This paper seeks to solve power flow equations using a combination of an attention-based graph neural network architecture with an iterative self-corrective scheme. The proposed methodology is evaluated on small power grids with up to 300 buses, and is shown to yield accurate voltage predictions and modest speedups compared to a classical CPU-based power flow solver.

Overall, I find the paper hard to follow, due to a lack of mathematical clarity, multiple typos in the document, and several missing details regarding the experiments.
The numerical experiments should be drastically expanded (using power grids with several thousands of buses) to demonstrate the relevance of the proposed approach. In addition, performance evaluation should be more robust (e.g. by reporting performance variability and additional accuracy metrics)

**Strengths:**

* The paper introduces an attention-based mechanism to propagate information efficiently across the power network graph, which can potentially mitigate the fact that power network graphs have high diameter.
* The "self-ensembling" mechanism adds an iterative correction scheme to mitigate potential prediction errors.

**Weaknesses:**

* The paper does not state the mathematical problem it is trying to solve. Namely, Section 3.1 only includes an English description, it would be beneficial for non-power systems experts to include a complete mathematical statement (i.e., definitions + math equations) of the problem to be solved.
* The following works are relevant for the literature review:
  * [Neural networks for power flow: Graph neural solver](https://doi.org/10.1016/j.epsr.2020.106547): one of the first works on graph neural networks for solving power flow
  * [https://doi.org/10.1016/j.epsr.2024.110738](https://doi.org/10.1016/j.epsr.2024.110738): proposes topology augmentation to mitigate the limitations of message-passing in sparse power network graphs.
* The proposed approach relies on a supervised learning loss for training, which requires labeled data that can be expensive to compute (especially for large-scale grids)
* Numerical results
  * The numerical experiments consider very small, artificial networks with artificially-sampled data. Experiments should consider industry-size networks (i.e. several thousands of buses)
  * The performance evaluation focuses on errors on voltage magnitude and angles, which does not capture potential mismatches in reactive power injections at PV buses
  * Numerical results do not report computing times (i.e. labeling, training and inference times) for ML baselines
  * The ablation studies do not consider removing the "fusion" component in itself.
* There are several typos in the manuscript, e.g.: non-matched closing parenthesis `)` on line 202, subscript `gt` at line 204. Please avoid abbreviations like "the net"

**Questions:**

* Please comment on the distributed slack setting, where the active power mismatch is distributed across the entire network. This more closely matches the reality of transmission grids, where multiple units adjust their output to ensure active power balance.
* Please explain the difference between parameters $\theta_{s}$ and $\theta_{t}$: they seem to refer the PPFNet parameters (l 233) and the "ensembling model" (l.247), but the latter is updated using the former.
* What does PPFNet actually output? Figure 1a suggests that it outputs $V_{m}, V_{a}$ (from which one evaluates the loss function), but Figure 2b suggests that it outputs a correction $\Delta V_{m}, \Delta V_{a}$?
* Please provide some measure of performance variance (e.g. standard deviation of loss across 10 different random seeds) in numerical results
* Please provide a comparison of the self-ensembling mechanism against a gradient-descent based mechanism whereby the initial prediction is used to warm-start a gradient-based algorithm that adjust $V_{m}, V_{a}$ to minimize the power imbalance. I expect the corresponding per-iteration cost to be substantially cheaper than the self-ensembling model.
* Please provide more details regarding the pypower implementation, especially regarding the use of parallelization using multiple CPU cores. The computational comparison should take into account the fact that an A100 GPU is about 100x more expensive than a single CPU core (using cloud computing price estimates of ~1c/hr per CPU vs ~1\$/hr per A100 GPU)
* Since the ML models are evaluated on GPUs, a GPU-enabled batch power flow solver like ExaPF is a more relevant baseline (see https://exanauts.github.io/ExaPF.jl/stable/)

---

### Official Review · Reviewer_WNVQ · 2025-11-08

**Soundness:** 2
**Presentation:** 2
**Contribution:** 2
**Rating:** 2
**Confidence:** 4

**Summary:**

This paper proposes a data-driven power flow analysis (PFA) method. The authors designed a learning mechanism embedded in the iterative process, along with a neural network architecture that accounts for the influence of the slack bus. Finally, several small standard IEEE cases are used to verify the proposed method’s advantage in terms of PFA accuracy.

**Strengths:**

## Strengths

- This paper embeds the neural network into the iterative process of the traditional PFA method, which differs from most existing end-to-end learning methods and makes the model’s training more conservative.
- This paper distinguishes between different types of buses in PFA in order to learn representations that are more effective for PFA.

**Weaknesses:**

## Weakness

- The term “power flow estimation” can be easily misunderstood, and from the perspective of power systems, the motivation of this paper is unclear. PFA has already been extensively studied, and even in large-scale systems with tens of thousands of buses, fast batch PFA can be achieved through high-performance computing techniques. So, why is a data-driven PFA necessary?
- The description of the correlation between the two components in SenseFlow is not very clear; they appear to be two components that could operate completely independently.
- The test cases are too small in scale and do not include renewable energy sources, making it difficult to demonstrate the practicality of the proposed method.

**Questions:**

## Questions

- In practice, there is often more than one slack bus in a power system. How does the proposed method handle this situation?
- How does the proposed method perform on samples where some bus voltages or line flows are at the security constraint boundary? For these samples, the PFA tool should accurately determine whether the operational variables exceed the boundary, rather than relying solely on the MSE evaluation.
- PFA is usually treated as a post-processing task of state estimation. Why do the authors consider the reactive power at buses in PFA to be unmeasurable, when it should be determinable through state estimation?
- During the iterative process, the calculation of deviations in bus injection powers depends on accurate line parameters. If the line parameters are inaccurate or missing, how significant would the impact be on the learning and inference of the entire model?
- The essence of the proposed method is to replace the computation of the Jacobian matrix. How significant is the impact of the accumulated prediction errors from the neural network on the results?
- Some arrows in Figure 3 are disconnected, please correct them.
- In the generation of the dataset, what do the line features refer to? Please clarify.

---

### Official Review · Reviewer_VQA8 · 2025-11-10

**Soundness:** 3
**Presentation:** 3
**Contribution:** 2
**Rating:** 4
**Confidence:** 5

**Summary:**

This paper proposes SenseFlow, a physics-informed graph neural network framework for estimating power flow in electrical grids, combining system topology with physical constraints. The approach iteratively refines predictions using self-ensembling and dual loss functions to enforce physical feasibility.

**Strengths:**

1. The problem addressed is timely and relevant to the increasing complexity of modern power systems.
2. The presentation is clear, and the technical pipeline is well-structured.
3. The proposed architecture and iterative refinement are technically valid and leverage recent advances in graph neural networks.

**Weaknesses:**

1. The paper’s core contribution is not clearly distinguished from existing machine learning approaches to power flow problems, particularly end-to-end methods that enforce physical correctness.
2. There is no theoretical analysis or guarantee provided for the prediction quality, convergence speed, or robustness of the proposed method.
3. The approach does not explicitly address the issue that power flow equations can have multiple feasible solutions, raising concerns about the uniqueness and consistency of the learned mapping.
4. The connection between the proposed method and the broader literature on learning optimal power flow (OPF) problems is insufficiently discussed, leaving unclear whether OPF methods could handle the considered problem as well or better.

**Questions:**

1. What is the main technical innovation of SenseFlow compared to prior ML-based power flow solvers, especially those using end-to-end learning with physical projection?
2. Does the method offer any formal guarantees regarding convergence, feasibility, or error bounds for the predicted solutions?
3. How does the model address the non-uniqueness of power flow solutions during training and inference, and is there any mechanism to ensure consistency or physical meaningfulness in the solutions it learns?
4. Could existing OPF learning methods be applied to the problem considered in this paper, and if so, what is the unique advantage of SenseFlow? Is the proposed approach extendable to OPF tasks, or are there limitations?
5. How well does the model generalize to unseen grid topologies or operating conditions?
6. What is the computational cost of the iterative self-ensembling process, and is the method scalable to large systems?
7. Can the model’s predictions be interpreted or trusted by system operators, and is there a way to quantify uncertainty in the outputs?
8. Have ablation studies or sensitivity analyses been carried out to assess the contribution of each architectural component and the impact of hyperparameter choices?

---

### Official Review · Reviewer_bxbM · 2025-11-11

**Soundness:** 2
**Presentation:** 2
**Contribution:** 2
**Rating:** 2
**Confidence:** 5

**Summary:**

The paper proposes SenseFlow, a GNN-based iterative estimator for AC power-flow state (voltage magnitudes and angles). The model contains (i) a Physics-informed Power Flow Network with Virtual Node Attention  and Slack-Gated Feed-Forward modules to increase global communication and emphasize the Slack node, and (ii) a Self-Ensembling Iterative Estimation training/inference loop that EMA-ensembles parameters across iterations. Experiments on multiple IEEE system datasets with perturbed injections/branches and random line outages are reported by authors. SenseFlow shows much smaller RMSE than competing GNNs and claims advantages (estimation under missing Q, faster N-2 contingency throughput vs PyPower, and scalability).

**Strengths:**

The paper makes an interesting attempt to tailor graph attention mechanisms specifically for power flow by emphasizing the role of the slack bus and physical dependencies. Also, the iterative, self-ensembling refinement idea shows awareness of the classical Gauss–Seidel/Newton–Raphson structure, which is conceptually appealing.

**Weaknesses:**

The authors seem to have missed the fundamental point of the power flow formulation — bus types (PQ, PV, slack) are not intrinsic physical categories but numerical conveniences to make the equations solvable. Also the slack bus itself merely absorbs system mismatch, not a source of “global influence” that must be learned. In reality, system works in distributed slack fashion (Dhople et. al. ) where all generators shares a mismatch of power, compared to schedule dispatch. Also, existing GNNs already capture network sparsity and coupling through the incidence or admittance matrix, so the claimed need for a special slack-aware attention is conceptually weak. Note that Gaussian Process–based models with physically informed kernels have already demonstrated how both local and global relationships in power systems can be modeled effectively and with far fewer samples (Bandong et.al., Pareek et.al.). Here, the network requires hundreds of thousands of samples  which raises concerns about scalability and true learning of physical structure.

**Major Concerns:**

1. **Slack bus understanding and discussion:**
The SGF block assumes a single slack bus and distributes its features globally. In practice, this is not physically meaningful under uncertainty — real systems use distributed slack where all generators share imbalance. This has been studied extensively (e.g., Dhople et al.). The current approach ignores this and risks non-physical generalization. Further, I am not sure how can you make NRLF converge for 50-150% uncertainty in load with single slack bus. A slack bus in these systems with this much load variation (specially when you hit 150%) but be supplying way beyond its capacity, which is far away from real situations and thus biasing the data towards slack-bus attention artificially.,

2. *Related work (GP, distributed slack, classical solvers):*  Gaussian process–based power flow/state estimation works;Distributed slack formulations; Modern scalable solvers such as nonlinear algebraic multigrid and HELM are not discussed. See _Machine Learning for Solving Power Flow Equations_ at https://energy.hosting.acm.org/wiki/index.php/ML_OPF_wiki . This makes the contribution positioning unclear.

3. **Experimental issues:** Extremely large training sets (up to 500k samples) with no analysis of data efficiency. Think like these, why should I not just solve power flow on a particular instance if I need to solve these many to train. Also no comparison to modern numerical solvers; only older GNN baselines and PyPower runtime is given + No report of FLOPs, wall-clock, or inference cost relative to solver accuracy.

4. **Lack of theoretical grounding**
The iterative SeIter process lacks any convergence or stability analysis. It is unclear whether it always improves residuals or can diverge.


**Minor Concerns**

– Missing hyperparameters: number of layers, K, attention heads, etc.
– Equations (7–12) need clearer explanation or pseudocode.
– Several typos and unclear notation (e.g., PV vs PQ example).

Dhople et. al. : Reexamining the Distributed Slack Bus, IEEE Transactions on Power Systems ( Volume: 35, Issue: 6, November 2020)
Bandong et.al. : https://arxiv.org/pdf/2505.15950

GP Works+Local Kernel + Topologies: Pareek et.al. Data-efficient strategies for probabilistic voltage envelopes under network contingencies, Sustainable Energy, Grids and Networks. (https://arxiv.org/abs/2310.00763)

**Questions:**

1. It’s not clear whether you’re solving power flow or state estimation — sometimes the paper reads like one, sometimes the other (especially when talking about missing Q or uncertain data). Can you clarify exactly what the task is and what inputs/outputs the model sees?

2. The slack bus treatment is physically confusing. The slack only absorbs system mismatch; it’s not a “global messenger.” Why is a special Slack-Gated module needed, and how would this extend to the more realistic distributed slack formulation where all generators share mismatch?

3. The model uses hundreds of thousands of samples for training — that’s a lot. How data-hungry is SenseFlow? Would performance degrade sharply with smaller datasets? The  out-of-distribution tests information is limited to N-1 or N-2 line removals. How much these lines were loaded on avg? Were the least or most loaded lines? How much voltage distribution changed once you remove these lines?

4. The SeIter refinement loop looks like a learned Gauss–Seidel update. Does it ever diverge or oscillate on difficult cases?

5. Power flow is a steady state problem. How are you relating it with stability?

---

### Note · Authors · 2026-01-12

I have read and agree with the venue's withdrawal policy on behalf of myself and my co-authors.